# Lost by Transcription: Fork Failures, Elevated Expression, and Clinical Consequences Related to Deletions in Metastatic Colorectal Cancer

**DOI:** 10.3390/ijms23095080

**Published:** 2022-05-03

**Authors:** Marcel Smid, Saskia M. Wilting, John W. M. Martens

**Affiliations:** Erasmus MC Cancer Institute, Department of Medical Oncology, University Medical Center Rotterdam, 3015 GD Rotterdam, The Netherlands; m.smid@erasmusmc.nl (M.S.); s.wilting@erasmusmc.nl (S.M.W.)

**Keywords:** metastatic colorectal cancer, structural variants, common fragile sites, platinum therapy, stalled replication fork

## Abstract

Among the structural variants observed in metastatic colorectal cancer (mCRC), deletions (DELs) show a size preference of ~10 kb–1 Mb and are often found in common fragile sites (CFSs). To gain more insight into the biology behind the occurrence of these specific DELs in mCRC, and their possible association with outcome, we here studied them in detail in metastatic lesions of 429 CRC patients using available whole-genome sequencing and corresponding RNA-seq data. Breakpoints of DELs within CFSs are significantly more often located between two consecutive replication origins compared to DELs outside CFSs. DELs are more frequently located at the midpoint of genes inside CFSs with duplications (DUPs) at the flanks of the genes. The median expression of genes inside CFSs was significantly higher than those of similarly-sized genes outside CFSs. Patients with high numbers of these specific DELs showed a shorter progression-free survival time on platinum-containing therapy. Taken together, we propose that the observed DEL/DUP patterns in expressed genes located in CFSs are consistent with a model of transcription-dependent double-fork failure, and, importantly, that the ability to overcome the resulting stalled replication forks decreases sensitivity to platinum-containing treatment, known to induce stalled replication forks as well. Therefore, we propose that our DEL score can be used as predictive biomarker for decreased sensitivity to platinum-containing treatment, which, upon validation, may augment future therapeutic choices.

## 1. Introduction

Large cohorts of primary cancers analyzed by whole-genome sequencing (WGS) were recently described [1], yielding a deluge of defects observed in cancer genomes. Most emphasis in these studies was on driver genes and mutational signatures [2,3,4], their disease-specific patterns, and their possible etiology. Building upon these shoulders, metastatic lesions of cancer patients have also been investigated in depth using WGS [5,6,7,8], describing enriched genetic aberrations and mutational signatures related to progression or to prior treatment. In metastatic colorectal cancer (mCRC), we previously detailed various somatic events and compared these to those found in primary CRC [9]. 

One result was that deletions in genes present in common fragile sites (CFS) were associated with response to therapy, and this current manuscript further investigates this. Large deletions, along with tandem duplications (DUPs), inversions, and translocations, are a class of structural variants (SVs) observed in most cancer types. In a primary pan-cancer study, these SVs have recently been extensively described, reporting high numbers of complex SVs, deletions, and tandem duplications in over 20 cancer types, including colorectal cancer. Other types of SVs are reported in virtually all cancer types, but at lower frequency. However, CFSs were only briefly discussed [10]. Among all cases, CFSs showed a peak DEL size of around 100 kb, and among the CRC cases (n = 52), the well-known CFS regions containing genes *FHIT* and *MACROD2* were most often identified (>40% of CRC cases).

CFSs are a decades-old observation [11], quickly gaining interest from the cancer field after reports of recurrent CFSs, with structural variation most often in the form of deletions [12,13,14]. Generally, CFSs are genomic regions that are late-replicating, AT-rich, often contain large genes, and showing tissue specific patterns (reviewed by [13]). In cancer, focal deletions are often associated with CFS; in a pan-cancer analysis [15] of 4934 cases, 70 recurrent focal deletions were reported, of which 22 were found in large genes and several genes known from CFS regions (*FHIT*, *WWOX*, *PDE4D*, *PARK2*). In terms of the mechanism driving the genome fragility of these regions, Wilson et al. distilled their observations in cell lines to a possible model involving a transcription-dependent double-fork failure (TrDoFF) [14]. In short, transcription of large genes possibly persists into the S-phase. This prevents firing of late (or dormant) replication origins that would otherwise be used to rescue stalled replication forks and thereby complete replication. In CFS, the stalled converging replication forks (i.e., a double-fork failure) create large, non-replicated regions which ultimately are resolved by deletion of the non-replicated DNA and duplications arising on the flanks, suggested to occur via template switching and microhomology-mediated break-induced replication (MMBIR) or alternative end joining [14,16]. 

We here aimed to have a detailed look into the role of the DELs in metastatic lesions of 429 mCRC patients, using WGS and corresponding RNA-sequencing data. We evaluated whether the instability at CFSs described in primary CRC persists in metastatic disease. We specifically studied the localization and size of DELs and DUPs, and included expression of the genes affected by these SVs. Lastly, we investigated the potential clinical consequences of these characteristics.

## 2. Results

### 2.1. Position and Size of Chromosomal Deletions in Metastatic mCRC

We previously described [9] density profiles of SVs (Figure 1a) in a large cohort of 429 metastases of colorectal cancer (mCRC) and noticed that the deletions (DELs) profile showed a peak at ~10 kb–1 Mb, which is not observed in metastatic prostate or breast cancer (Figure 1b). Furthermore, these DELs were not distributed randomly across the genome (Figure 1c), but instead showed a clear preference for specific chromosomal locations. Using a threshold of at least 150 events, we established 13 hotspot locations (see Appendix A and Table 1) which were covering, or, for all but one, located nearby known common fragile sites (CFSs) [17]. For all hotspot locations, many samples had more than a single DEL in the region, often overlapping (Table 1 and an example in Appendix A). We evaluated the sequence around the breakpoints of the DELs but were unable to find enriched motifs, besides the sequence logos showing AT-rich sequences (Appendix A), which is a well-known feature of CFSs (reviewed by [13]). In summary, the mCRC genome contains DELs of a specific size at specific chromosomal locations, linked to well-known CFSs.

We next used previously reported locations of several potentially relevant genomic features to investigate the reason behind the preference for the DEL size, including topologically associated domains (TADs) [18] and sites where initiation of replication was found to occur (Ini-seq peaks) [19]. TADs are structural features of genomic organization, where regions bordering TAD boundaries are thought to regulate gene expression. Both TADs and Ini-seq peaks often flanked the regions were the DELs were found. See for example *RBFOX1* in Figure 2a. Next, we systematically evaluated whether both breakpoints of a DEL (size 10 kb–1 Mb) were located between consecutive Ini-seq peaks or TADs (Figure 2b). Outside the 13 hotspot regions, 68% of DELs (6488 out of 9553) were located between consecutive Ini-seq peaks, whereas inside the hotspot regions, this was the case for 92% of DELs (4836 out of 5237, chi-sq *p* = 1.29 × 10^−246^, odds ratio 5.697). For TADs, 84% vs. 90% of DELs were in between TADs outside and inside hotspot regions, respectively (chi-square *p* = 5.96 × 10^−25^), and with an odds ratio of 1.738, having a much smaller effect size compared to Ini-seq peaks. A multivariable logistic regression of DELs within Ini-seq peaks and TADs with being inside or outside hotspot region as outcome variable showed odds ratios of 5.43 and 1.14 (*p*-values 1.33 × 10^−189^ and 0.025), respectively, indicating that Ini-seq peaks have an exceedingly larger role in bordering DELs in the hotspot regions. 

Since it is well known that CFS regions are late-replicating, it seems reasonable that replication origins (identified by the Ini-seq peaks) are of importance for SVs therein, and it is likely that the observed preference of DEL sizes in mCRC are restricted by the location of the bordering Ini-seq peaks. A histogram of the distances between all consecutive Ini-seq peaks (Figure 2c) showed a median of 24.9 kb (95% confidence interval (CI) 24.2–25.7 kb) with 95% of observations below 425 kb. However, the median distance between Ini-seq peaks at the hotspot regions is 1647 kb, which would be the upper bound if bordering replication origins determine the DELs size in CFS. Furthermore, assuming that converging replication forks starting from the bordering Ini-seq peaks are arrested at some point followed by deletion of the intervening region, DELs are expected to concentrate around the midpoint of the region. This assumption matched with the observations in our cohort; DELs within the 13 hotspot regions were much more localized toward the midpoint of the bordering Ini-seq peaks, compared to the more random localization of DELs outside the hotspot regions (Figure 2d). 

### 2.2. Deletion and Duplication Patterns Concur with the Transcription-Dependent Double-Fork Failure Model

These observations suggest that the DELs may be a consequence of stalled replication forks, the reason for their stalling likely linked to the particular genomic milieu. A potential solution for these stalled replication forks was proposed by the transcription-dependent double-fork failure (TrDoFF) model [13,14], in which resolution of these stalled replication forks leads to DELs centering around the midpoint of the gene and duplication (DUP) at the flanks of the region. In support of this model, we find that DUPs (Figure 3a,b) are indeed also enriched at the same loci as DELs in mCRC, although with a much lower prevalence and of smaller size (Appendix A). The exact location of DUPs and DELs is exemplified in Figure 3c, showing the events at the FHIT locus. Systematic evaluation of the genes located in the hotspot CFS regions, taking the transcription direction into account, showed that DELs are indeed centered around the midpoint of the genes, with DUPs more present at the flanks of the region (Figure 3d).

Additional support for an actual role of TrDoFF in mCRC is provided, employing available gene expression data (Figure 4). For example, *NAALADL2* (located in FRA3C, genome size 951 kb), one of the genes with many DELs, was expressed and located between Ini-seq peaks, while the neighboring *NLGN1* gene, also large (size 898 kb) and located between Ini-seq peaks, but having no accompanying DELs, was not expressed. We verified this observation genome-wide by selecting large genes (>500 kb) within and outside the 13 CFS regions but positioned between Ini-seq peaks (*n* = 147 genes). We showed that the median gene expression level of the genes inside the hotspot regions was significantly higher (Mann–Whitney *p* = 0.0013) than that of the genes outside hotspot regions (Figure 4b).

Of note, not all 13 hotspot regions contained a single large gene at the location where DELs were observed (Table 1 and Appendix A). For the hotspot located on chromosome (chr) X, multiple genes are located within consecutive Ini-seq peaks, with DELs and DUPs, respectively, centering and flanking two genes (*PUDP* and *STS*). The hotspot region on chr 8 has no gene annotated to the region where DELs/DUPs are located. An uncharacterized noncoding RNA LOC105375631 is located nearby, and visual inspection of RNA-seq bam files shows some mapped reads there (data not shown).

In summary, the specific DEL size and DEL/DUP position seen in mCRC appeared to be guided by two observations: (1) the origins of replication flanking the CFS regions and (2) active transcription. The TrDoFF model proposes that the combination of late replication and transcription of large genes gives rise to double-fork failure. The disentanglement of that complex yields DELs and DUPs at specific locations which precisely fit with the observations presented here. 

### 2.3. Cells with DELs in CFS Retain Transcription of the Locus

For samples with available RNA-seq data, we searched for sequence reads that corroborate the DEL reported at DNA level in that sample, by identifying reads that cross the junction created by the DEL. In total, 305 sequence reads were identified in 44 samples (Figure 5). Though often only a single read showed the presence of the DEL, in all 13 CFS regions multiple samples showed evidence of such reads. Each of these sequence-reads were uniquely aligned to the genome, with part of the read mapping immediately before the 5′ position of a DEL with the remaining part of the sequence mapping directly after the 3′ position of the DEL. Appendix A shows an example of these in *WWOX* in a single sample. 

### 2.4. Clinical Implications

We considered the number of DELs (10 kb–1 Mb) within the 13 hotspot regions as a biomarker for the ability to circumvent stalled replication forks. We hypothesized that such a mechanism would act on any double-fork failure, not necessarily transcription-dependent or just those in CFS regions. Since we previously observed that loss of some of the genes in hotspot regions (e.g., *WWOX*, *PARK2*) were associated with response to platinum-containing therapy [9], we reasoned that the mechanism that resolves stalled replication forks may also be able to resolve platinum-caused cross-links in DNA that potentially lead to stalled forks as well. We considered samples >20 DELs in the 13 hotspot regions (see Appendix A) as resolved stalled fork positive (RSF+).

In total, 118 patients received platinum-containing therapy after biopsy (14 of whom are RSF+, see Table 2), and associating RSF groups with progression-free survival (PFS) showed that this was significantly shorter in RSF+ patients (Figure 6, left panel, log rank *p* = 0.0001). A subset of patients that were given platinum-containing therapy after biopsy were also previously treated with this agent (i.e., prior to biopsy of the metastasis that was sequenced), a clinical decision often taken when the patient had a relatively good response (defined as the interval between last platinum-containing therapy and time of progression >6 months). A multivariable analysis to test the association of RSF status, prior platinum therapy, and the number of prior-treatment lines with PFS on subsequent platinum-containing therapy showed a relative hazard ratio (HR) for RSF+ of 3.01 (*p* = 0.0029, 95% CI 1.46–6.21), HR = 0.83 (*p* = 0.10, 95% CI 0.20–1.15) for prior platinum therapy and an HR of 1.58 (*p* = 0.038, 95% CI 1.03–2.42) for the number of prior-treatment lines.

Lastly, patients who did not receive any prior systemic treatment before biopsy also showed a significantly shorter PFS on platinum-containing therapy in RSF+ patients (Figure 6, right panel, log rank *p* = 0.0007). The group of patients with any prior treatment (platinum or other) was too small to perform a statistically meaningful analysis, with only two RSF+ patients in either prior containing platinum or prior other. These four patients did show a similar trend with a PFS < 250 days.

## 3. Discussion

We here describe an in-depth analysis of DELs in metastatic lesions of 429 CRC patients. The density profile of DELs in mCRC is distinct from those in metastatic breast- and prostate-cancer but bears resemblance to a profile seen in, e.g., primary esophagus and pancreas cancer [10], with the top affected CFS loci and their connected genes (*FHIT*, *MACROD2* and *WWOX*) also among the reported regions here. Using their pan-cancer set of primary tumors, Li et al. reported 18 CFS regions [10], of which 10 out of our 13 regions overlapped. The observed DEL and DUP patterns fully support the TrDoFF mechanism, whereby these SVs are generated in CFS [14]. In short, the model, which is based on experimental data in in vitro model systems, suggests that in large genes during replication stress, transcription is still active in the S-phase, precluding activation of dormant replication origins to resolve stalled replication forks. As a result, in late-replicating regions that are actively transcribed, converging stalled replication forks cannot be rescued. To resolve these, template switching across both stalled forks gives rise to DELs while occasional re-initiation of replication results in DUPs near the origins of replication. Another frequently observed aspect of the phenotype is that within a single tumor lesion, multiple, often overlapping, DELs are present within the same gene/CFS. This is likely due to clonal heterogeneity where subpopulations of clonal cells each have their own specific deletion. The RNA-seq data agree with this notion, since sequence reads can positively identify multiple and overlapping DELs within the same sample. We hypothesize that in different cells of a tumor, the transcription complex is probably stochastically located at different positions in the gene, stalling the converging replication forks at different locations, resulting in distinct DELs. The fact that CFS regions are not all entirely lost in CRC during progression into these metastatic lesions implies that the TrDoFF mechanism is not active in all cancer cells or in every cell division. The events we do observe are those derived from expansion of individual cells in which the initial, and specific, event occurred.

To the best of our knowledge, we are the first to match observations in a large cohort of metastatic CRC patients with four essential features of the TrDoFF model: (i) sparsity of replication origins, (ii) transcription, and the (iii) presence and (iv) position of DELs and DUPs. For the latter three features, we were able to use in-house data only; for the sparsity of origins we additionally required public data (Ini-seq peaks) which were based on a replication origin mapping technique whereby newly replicated DNA is labeled and sequenced [19]. The position of these Ini-seq peaks show that DELs are enriched between two consecutive Ini-seq peaks in CFS regions, which would be in line with CFS regions having no active origin of replication to rescue any stalled replication forks converging from the flanks of the CFS regions. Of note, the locations of the Ini-seq peaks in CFS regions thus provide an upper bound for the size of the DELs observed in mCRC. Lastly, it is important to realize, as also addressed by the original authors [13,14], that a key event in the formation of DELs in CFS is that transcription appears as the reason for the absence of an active replication origin that could rescue stalled replication forks. There may be underlying causes other than transcription for the inactivation of origins, or other mechanisms responsible for double-fork failure or incomplete replication in general. 

Indirect clues for other contributing factors come from the observation that patients that received platinum-containing treatment prior to removal of the metastatic biopsy showed a significantly higher number of DELs in CFS. Platinum compounds induce crosslinks between two bases, either GG or AG, in DNA [20,21] while interstrand crosslinks potentially block transcription and replication [22]. We argue that in mCRC patients the mechanism that is used to resolve the transcription-dependent double-fork failure may also act on stalled forks caused by platinum crosslinks. The observation that PFS on platinum-containing therapy is significantly shorter in patients with a high number of DELs (RSF+) is in line with this hypothesis. The clinical decision to administer platinum-containing therapy may be augmented by only including RSF- patients, since the benefit of platinum-containing therapy for RSF+ patients seems limited. Prospective trials where patients are properly stratified during inclusion are required to evaluate the true clinical applicability of the RSF status.

In conclusion, we here present data that show that CFSs are still present in mCRC, providing an explanation for why DELs observed in mCRC are of a certain size, and that the observations agree with a model of transcription-dependent double-fork failure as modus operandi for DELs (and DUPs) in CFS. This ability to resolve stalled replication forks is associated with resistance to platinum-based therapy, which may provide new prospects for clinical decision making.

## 4. Materials and Methods

### 4.1. Study Cohort

The cohort consists of 429 metastatic biopsies taken from colorectal cancer patients gathered as part of the Center for Personalized Cancer Treatment (CPCT) consortium (CPCT-02 Biopsy Protocol, ClinicalTrial.gov no. NCT01855477, 16 May 2013), which was approved by the medical ethics committee of the University Medical Center Utrecht, the Netherlands [6]. All patients gave explicit consent for whole-genome sequencing and data sharing for cancer research purposes. Additional characteristics were previously described [9], but in short, whole-genome sequencing (WGS) of paired tumor/normal was performed in all cases. Raw sequencing data were processed using bcl2fastq (versions 2.17 to 2.20) and mapped to the human reference genome GRCh37 using BWA-mem v0.7.Sa (https://github.com/lh3/bwa). Of all tumor biopsies, 98% had a coverage of at least 30 × (95% with >60 × coverage), whereas for the normal blood, 98% had >10× coverage and 94% >20 × coverage. Structural variants were called using GRIDSS v1.8.0 [23]. RNA-sequencing data were available for 343 samples and RNA-seq was performed as described previously [9] with gene expression data (adjusted TPM) obtained using Isofox (https://github.com/hartwigmedical/hmftools/tree/master/isofox). Metastatic patients starting any new systemic treatment were included, and the particular treatment given to participating patients was decided by the local clinicians. The types and lines of systemic treatment given before the biopsy was taken were recorded; in total, 241 patients received a platinum-containing therapy before biopsy, mostly Oxaliplatin in combination with Capecitabine (*n* = 231), while 124 patients were treatment-naïve at the time of biopsy. Furthermore, response to treatment after biopsy was evaluated; 118 patients received a platinum-containing therapy after the biopsy was taken with available progression-free survival data (again, mostly (*n* = 85) in the form of a combination of Oxaliplatin + Capecitabine with/without Bevacizumab). 

### 4.2. Statistics

For categorical data, a Pearson’s chi-squared test was used while continuous variables were evaluated using a Mann–Whitney U-test (MWU). All statistical tests were two-sided and considered statistically significant when *p* < 0.05. Stata 13.0 (StataCorp, College Station, TX, USA) and R (v3.6.0) were used for the statistical analyses. R package “mcp” (https://github.com/lindeloev/mcp/) was used to perform piecewise linear regression in order to find change points in the number of deletions per sample. The midpoint between the first two changepoints was selected to indicate samples with a high nr of DELs (>20). KaryoploteR [24] was used for visualizations. The sequence logo was generated via WebLogo [25].

## Figures and Tables

**Figure 1 ijms-23-05080-f001:**
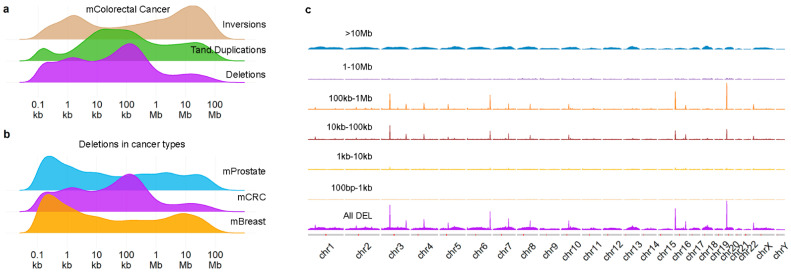
Deletions in mCRC show spatial and size preference. Density plots of structural variant in mCRC (**a**) and of deletions in 3 different metastatic cancers (**b**); (**c**) shows density plots by the indicated sizes across the genome. Figure 1a reproduced from Mendelaar et al., Nat Commun 2021, 12:574 (CC BY 4.0, https://creativecommons.org/licenses/by/4.0/, accessed on 1 December 2021) with modified colors.

**Figure 2 ijms-23-05080-f002:**
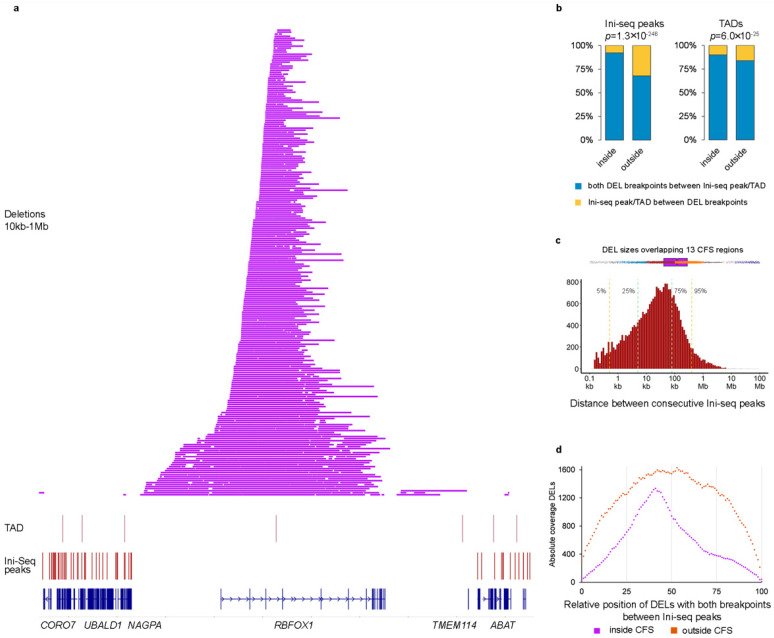
Origins of replication determine size of deletions in CFS. (**a**) Chromosome 16 region containing *RBFOX1*. Indicated are (bottom to top) the locations of genes, Ini-seq peaks, TADs, and DELs between 10 kb–1 Mb. (**b**) Bar graph showing enrichment of Ini-seq peaks/TADs inside CFS regions. *p*-values via Mann–Whitney U-test. (**c**) Top horizontal boxplot shows the size of all DELs within or overlapping the 13 CFS regions. DELs are colored by size according to the groups in Figure 2c. Bottom histogram shows the distribution of distance between 2 consecutive Ini-seq peaks. Vertical lines show the indicated percentiles. (**d**) Relative position of DELs between consecutive Ini-seq peaks.

**Figure 3 ijms-23-05080-f003:**
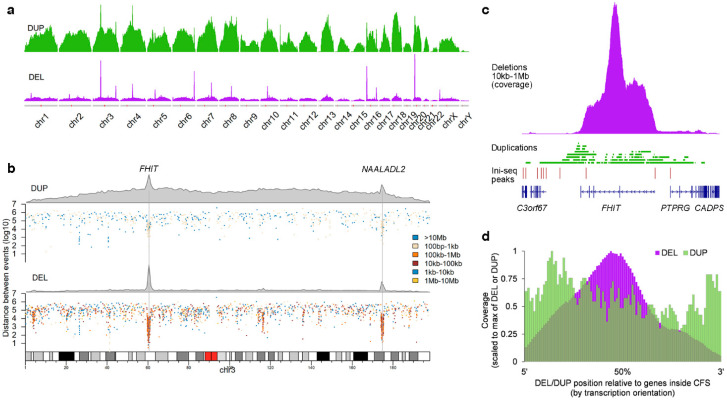
Spatial and size preference for duplications in mCRC. (**a**) Density plots of DELs and DUPs in mCRC. Each track is scaled to the maximum density within that track. (**b**) DELs and DUPs in chromosome 3, each showing a density plot and a rainfall plot to indicate sizes. (**c**) More detailed DEL/DUP locations for the *FHIT* gene. (**d**) Relative position of DELs/DUPs in hotspot CFS regions, corrected for transcription orientation of the genes.

**Figure 4 ijms-23-05080-f004:**
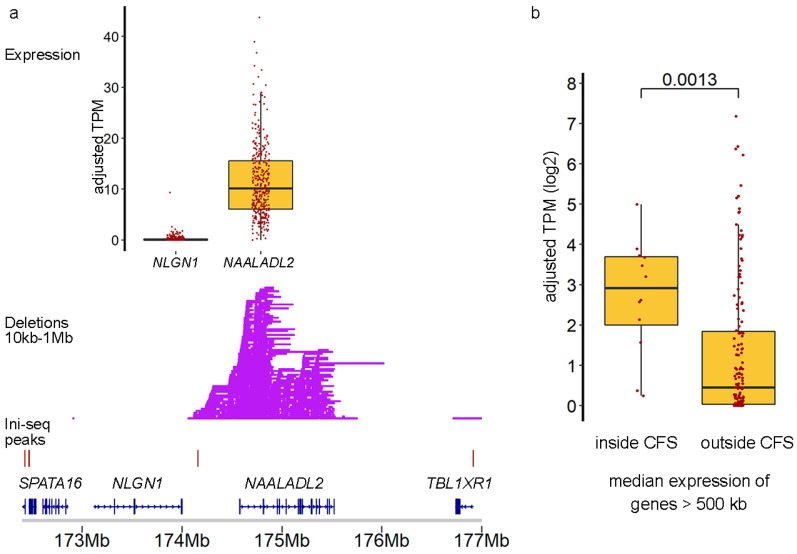
Transcription drives deletions in CFS regions. (**a**) Detailed plot of the *NAALADL2* and *NLGN1* genes, both large genes that locate between consecutive Ini-seq peaks, showing DELs occurring only in the expressed *NAALADL2* gene. (**b**) Boxplot of median expression levels of genes inside CFS vs. large (>500 kb) genes outside CFS. *p*-value via Mann–Whitney U-test.

**Figure 5 ijms-23-05080-f005:**
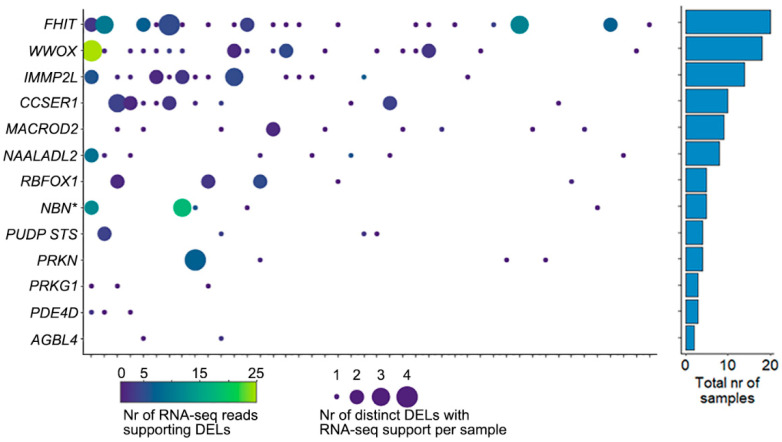
RNA-seq reads supporting DELs. The left panel shows the number of RNA-seq reads that support a DEL in one of the CFS regions. X-axis shows 44 samples, y-axis the 13 CFS regions (* indicates that *NBN* is the nearest gene). The color of the bubble indicates the total number of reads observed for all DELs in that sample/CFS region, and the size of the bubble indicates the total number of DELs with supporting RNA-seq reads in that sample/CFS region. The right panel shows the total number of samples that have RNA-seq support of DELs in the CFS region.

**Figure 6 ijms-23-05080-f006:**
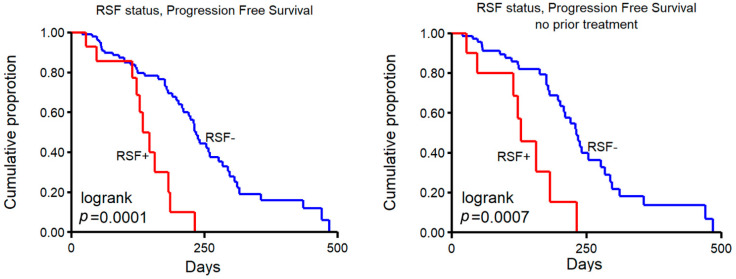
Number of DELs and progression-free survival. Kaplan–Meier progression-free survival curves of patients treated with platinum-containing therapy. Samples were grouped in those >20 DELs, labeled as resolved stalled fork positive (RSF+) vs. <20 (RSF-). Left panel shows all patients, and right panel shows patients with no prior therapy.

**Table 1 ijms-23-05080-t001:** Regions with at least 150 DELs in mCRC.

Chromosome	Start Position of First DEL	End Position of Last DEL	Gene	Fragile Site (HumCFS)	Nr. of Samples with >1 DEL
1	48435860	50548351	*AGBL4*	~0.6 Mb 5′ of FRA1B	34
3	59572890	61536465	*FHIT*	FRA3B	168
3	174058837	175755903	*NAALADL2*	~6.7 Mb 5′ of FRA3C	80
4	91027502	92737250	*CCSER1*	FRA4E	85
5	58229103	59931219	*PDE4D*	FRA5H	55
6	161666892	163416771	*PRKN*	FRA6E	125
7	109283013	111357245	*IMMP2L*	FRA7K	76
8	89804552	91152321	*NBN* *	~3.2 Mb 5′ of FRA8B	36
10	52569645	54085262	*PRKG1*	~0.6 Mb 5′ of FRA10C	40
16	5690370	7815010	*RBFOX1*	~7 Mb 5′ of FRA16A	173
16	78099748	79260329	*WWOX*	FRA16D	61
20	13921064	16054649	*MACROD2*	~1.8 Mb 5′ of FRA20A	207
X	6511238	7782627	*PUDP STS*	~21.9 Mb 5′ of FRAXB	59

* Nearest gene. HumCFS is data from [17]. Coordinates by hg19 reference.

**Table 2 ijms-23-05080-t002:** Patients treated with platinum-containing therapy.

Prior Treatment	Total	RSF+	RSF-
All	118	14	104

Platinum-containing	18	2	16
Other	19	2	17
None	81	10	71

## Data Availability

All data (WGS, RNA-seq, and clinical data) used in this study are made available by the Hartwig Medical Foundation (Dutch non-profit biobank organization) after signing a license agreement stating that data cannot be made publicly available via third party organizations. Therefore, the data are available under data request code DR-058 and can be requested by contacting the Hartwig Medical Foundation (https://www.hartwigmedicalfoundation.nl/applying-for-data/).

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
