# Peer review of "Lost by Transcription: Fork Failures, Elevated Expression, and Clinical Consequences Related to Deletions in Metastatic Colorectal Cancer"

_ijms, 2022, doi:10.3390/ijms23095080_

Round 1

Reviewer 1 Report

Martens et al., has summarized the “association of fork failures and deletions to metastatic colorectal cancer”. As deletions in metastatic colorectal cancer show a size preference of ~10 Kb-1 Mb and are often found in common fragile sites (CFSs), the author aimed to understand the reason behind this phenomenon and their possible correlation with patient outcome.

Further, the author suggested that Deletion score can be used as a predictive biomarker for determining the platinum-containing treatment sensitivity in metastatic colorectal cancer. The authors do a good job in explaining the bioinformatics-related analysis and results. There were a few grammatical errors but otherwise it is well written.

Author Response

Response:

We thank the reviewer for the assessment. We have reviewed the text for grammatical errors and corrected these.

Reviewer 2 Report

In this study, the authors evaluated the role of the DELs in metastatic lesions of 429 mCRC patients, using WGS and corresponding RNA-sequencing data. They assessed whether the instability at CFSs described in primary CRC persists in metastatic disease. They were specifically interested in the localization and size of DELs and DUPs, and included expression of the genes affected by these SVs. The manuscript is straightforward, well written, and concise and has clear results, within the scope of a retrospective analysis. Definitely deserves to be published and is a valuable contribution to the “International Journal of Molecular Sciences”. Some minor comments need to be addressed before publication.

[1] “1. Introduction”, Page 1 of 11, Lines 41-42:

Large deletions, along with tandem duplications (DUPs), inversions and translocations, are a class of structural variants (SVs) observed in most cancer types.”.

At that point, the authors should provide an example. Please, report that alterations in DNA damage repair (DDR) genes are common in prostate cancer, through mutations or deletions. The incidence of germline mutations in DDR genes is 11–33% among men with metastatic castration resistant prostate cancer, significantly higher than that of localized disease. The commonest DDR aberration in prostate cancer is BRCA2, followed by CDK12, ATM, CHEK2, BRCA1, MSH2, FANCA, MLH1, and RAD51.

Recommended reference: Ghose A, et al. Genetic Aberrations of DNA Repair Pathways in Prostate Cancer: Translation to the Clinic. Int J Mol Sci. 2021;22(18):9783.

[2]3. Discussion”, Page 8 of 11, Lines 234-237:

In short, the model, that is based on experimental data in vitro model systems, suggests that in large genes during replication stress, transcription is still active in the S-phase, precluding activation of dormant replication origins to resolve stalled replication forks”.

During replication stress, cells undergo arrest to allow time for repair and re-entry into the cell cycle. It should be mentioned that BRCA1 and BRCA2 are responsible for the protection of stalled replication forks. In the absence of BRCA1/2, nucleases such as MRE11 and MUS81 attack stalled replication forks, leading to fork collapse and chromosomal abnormality.

Recommended reference: Shah S, et al. BRCA Mutations in Prostate Cancer: Assessment, Implications and Treatment Considerations. Int J Mol Sci. 2021;22(23):12628.

Author Response

point [1]

We thank the reviewer for this suggestion. The lines immediately following state that these SVs have been extensively described in a primairy pan-cancer study. This study contained 2658 cancers across 38 tumor types. An example singling out prostate cancer seems a bit out of place, especially since we are studying colorectal cancer. Furthermore, we are introducing and reflecting on known literature regarding the frequency and cancer-type specificity of SVs and CFSs, and not which genes are involved in (mis)repair of SVs or in generating SVs, or what mutations are found in these genes. Finally, the recommended reference does not report anything on structural variants, common fragile sites or large genomic deletions. So it is unclear why this reference should be used as an example of structural variants observed in most cancer types.

To provide a relevant example, we have added the following:

“Large deletions, along with tandem duplications (DUPs), inversions and translocations, are a class of structural variants (SVs) observed in most cancer types.

In a primary pan-cancer study, these SVs have recently been extensively described, reporting high numbers of complex SVs, deletions and tandem duplications in over 20 cancer types, including colorectal cancer. Other types of SVs are reported in virtually all cancer types, but at lower frequency. However, CFSs were only briefly discussed [10].  

point [2]

We thank the reviewer for this additional insight. Lines 234-237 state a short summary of a model of transcription dependent double fork failure that was described by Wilson et al. In that model, BRCA1/2 are not mentioned, so including these genes at that point does not reflect the model as described. More importanty, BRCA1 and BRCA2 are rarely mutated in colorectal cancer (~6% of cases in our cohort), so their role, if any, is extremely limited in protecting stalled replication forks in the disease type we are studying.  Lastly, and the most important, the process of generating the deletions in CFSs via the TrDoFF-model relies on explicit functional repair processes, where the unreplicated DNA between the stalled replication forks is deleted and the remaining DNA rejoined. The genes that drive this repair machinery should be wild-type, not mutated. Thus, in colorectal cancer in CFSs, the chromosomal abnormalities are plainly not generated via the process the reviewer outlines.

Because of these arguments, we did not change the text since there is no support for involvement of BRCA1/2, not from the in vitro model, nor from the observations in our colorectal data or the proposed working mechanism.

Reviewer 3 Report

First of all, I would like to thank the authors for this wondeful and high quality extensive work. I am surely not adequately knowledgeable to even appreciate the work to the extent it deserves.  I have no concern about the importance, accuracy or interpretation of the results.

However, for clear understanding or communicating to general readers, I would request the authors to give a simple or plain definition, description and method of identification of the events like TAD, Ini-seq peaks, RSF, double fork failure etc. Perhaps some of the supplementary figures should be brought in the main paper to give the audiance a better understanding.

Considering the fact that a significant proportion of the mCRC are still treated with conventional chemotherapy because of lack of proven targetted therapy, there is the potential clinical application of this research finding. Taking that into consideration, I strongly suggest to present the sequence of at least the top hit regions in the supplementary data so that the result can be reproduced by other groups. 

Author Response

Response:

We thank the reviewer for the favourable assessment. We have elaborated on the requested items at several locations in the manuscript, see below. Regarding the supplemental figures, we are already at the maximum allowed, but, more importantly, we have opted to not interrupt the flow of the results too much with the, in our eyes, really supplemental results.

Regarding the clinical application: we agree with the potential benefits, but we are unsure of the particular request. First, all regions that were investigated are listed in table 1, including the chromosomal positions. These regions are fairly large, sometimes megabases, and we think it would be quite impractical to include the sequence of the regions. The provided coordinates in table 1 can be used to look up the sequence of the human reference, since that is in the public domain. Furthermore, our RSF score is based on the number of deletions we observed in these regions; the exact locations of the deletions differ between samples, and again supplying the sequence of the deletions will not give other groups useful information to reproduce the results. This is because 1) it is about the number of deletions and 2) the exact breakpoints of the deletions are not fixed, but rather an outcome of a stochastic process, yielding deletions at different locations within the regions listed in table 1.

To summarize, for other groups to reproduce our results, they would need to measure the number of deletions found in the chromosomal regions listed in Table 1. Thus all relevant and required information is presented for reproducing the results.

Changes made to text

-Line 62: In CFS, the stalled converging replication forks (i.e. a double fork failure) create large…

-Line 98-99: TADs are structural features of genomic organization, where regions bordering TAD boundaries are thought to regulate gene expression.

-RSF was already defined in Line 205-206:

We considered samples >20 DELs in the 13 hotspot regions (see Supplementary figure 5) as Resolved Stalled Fork positive (RSF+).

-Line 257-260: Ini-seq peaks were defined in line 97-98, and it has been made more clear in Lines 260-263 how these Ini-seq peaks are detected:

..for the sparsity of origins we additionally required public data (Ini-seq peaks) which was based on a replication origin mapping technique whereby newly replicated DNA is labelled and sequenced [19].